# Impact of Preoperative Visceral Fat Area Measured by Bioelectrical Impedance Analysis on Clinical and Oncologic Outcomes of Colorectal Cancer

**DOI:** 10.3390/nu14193971

**Published:** 2022-09-24

**Authors:** Kyeong Eui Kim, Sung Uk Bae, Woon Kyung Jeong, Seong Kyu Baek

**Affiliations:** Department of Surgery, School of Medicine, Keimyung University Dongsan Medical Center, Daegu 42601, Korea

**Keywords:** colorectal cancer, nutritional assessment, bioelectrical impedance analysis, visceral fat area

## Abstract

Background: Some studies have shown that an increase in visceral fat is associated with postoperative clinical and oncologic outcomes. However, no studies have used bioelectrical impedance analysis (BIA) to determine the effects of visceral fat on the oncologic outcomes of colorectal cancer (CRC). This study aimed to investigate the impact of preoperative visceral fat area measured by bioelectrical impedance analysis on clinical and oncologic outcomes of colorectal cancer Methods: This study included 203 patients who underwent anthropometric measurements by BIA before surgical treatment for CRC between January 2016 and June 2020. Results: According to the cut-off level of VFA by receiver operating characteristic curve analysis, 85 (40.5%) patients had a low VFA, and 119 (59.5%) had a high VFA. Multivariate analysis found that preoperative CRP (hazard ratio (HR), 3.882; 95% confidence interval (CI), 1.001–15.051; *p* = 0.050) and nodal stage (HR, 7.996; 95% CI, 1.414–45.209; *p* = 0.019) were independent prognostic factors for overall survival, while sex (HR, 0.110; 95% CI, 0.013–0.905; *p* = 0.040), lymphovascular invasion (HR, 3.560; 95% CI, 1.098–11.544; *p* = 0.034), and VFA (HR, 4.263; 95% CI, 1.280–14.196; *p* = 0.040) were independent prognostic factors for disease-free survival (DFS). Conclusions: Preoperative VFA measured by BIA had no significant impact on postoperative clinical outcomes and was an independent prognostic factor for disease-free survival.

## 1. Introduction

Colorectal cancer (CRC) is the third most frequently diagnosed cancer and the second most common cause of mortality worldwide [1]. Identifying markers of disease recurrence and poor prognosis is crucial for the successful treatment of CRC patients and the development of novel therapeutic options [2,3,4]. According to the World Health Organization, 39% of adults aged ≥18 years are overweight, and 13% of adults are obese [5]. The relationship between cancer and body weight is now well-recognized and obesity is now a well-established risk factor for the development of colorectal cancer and is associated with an increase in cancer-related mortality [6]. The underlying mechanisms correlating obesity with CRC have not been completely elucidated, but sustained inflammatory signaling, chronic insulin resistance, adipokine dysregulation induced by adipose tissue macrophages, and hypoxic and angiogenic environments of obese adipose tissue with elevated circulating cytokines have been proposed as important factors for carcinogenesis [7].

Body mass index (BMI) is one of the most reliable anthropometric methods for detecting obesity [8]; however it has no bearing on the accumulation of adipose tissue, particularly intra-abdominal or visceral fat tissue [9]. Controversies exist regarding the correlation between visceral obesity and colon cancer outcomes. Some studies have shown that visceral obesity is associated with poorer clinical and oncologic outcomes, including longer hospital stays, higher morbidity within 30 days, longer operation times, more aggressive pathologic tumor features, and poorer survival rates [10,11]. However, other studies have reported that visceral obesity has a protective effect on overall survival compared to non-visceral obesity [12,13].

Analysis of body composition describes the proportions of fat, protein, and minerals in human bodies. Bioelectrical impedance analysis (BIA) is a noninvasive technique that is cost-effective and available at many healthcare services for nutritional assessment and anthropometric analysis, including percentages of fat, protein, body fluid, and minerals in human bodies. Previous studies have shown the relationships between body composition, including using skeletal muscle index, visceral fat, phase angle, and clinical and oncologic outcomes of CRC [14,15]. However, to date, no studies have investigated the effects of visceral fat on the clinical, pathological, and oncologic outcomes of CRC using BIA. Therefore, this study aimed to investigate the relationship between visceral fat area (VFA) and oncologic outcomes in CRC.

## 2. Materials and Methods

### 2.1. Ethical Considerations

The institutional review board at Keimyung University Hospital approved the protocol for the retrospective study (approval number: DSMC 202207015). Data were acquired and analyzed ethically, while the patients’ right to privacy was respected. The informed consent requirement was waived.

### 2.2. Patients and Data Collection

This study was approved by the Institutional Review Board of the Dongsan Medical Center (Daegu, Republic of Korea, IRB No. 2022-07-015). The need for informed consent was waived due to the retrospective nature of this study. Between January 2016 and June 2020, 204 patients who underwent laparoscopic surgery for colorectal cancer were included in the study group. Exclusion criteria included concurrent or prior malignancies, malignancies other than adenocarcinoma, familial adenomatous polyposis (FAP) or hereditary non-polyposis colorectal cancer (Figure 1).

### 2.3. Data Collection and Definitions

A prospectively maintained database and electronic medical records were searched to collect the data. Data on patient demographics, including age, sex, American Society of Anesthesiology (ASA) score, preoperative carcinoembryonic antigen (CEA), BMI, and location of the tumor and prognostic inflammatory factors including platelet-lymphocyte ratio (PLR), neutrophil-lymphocyte ratio (NLR), prognostic nutritional index (PNI), and pan-immune inflammation value (PIV) were collected retrospectively using electronic medical records. ASA score is the method to assess and communicate a patient’s pre-anesthesia medical co-morbidities. ASA I was defined as a normal healthy patient. With increasing ASA score, the severity of co-morbidity increases, and ASA III was defined as a patient with severe systemic disease [16]. Perioperative outcomes included operation time, time to gas out, sips of water, soft diet, hospital stay, morbidity within 30 days, and Clavien–Dindo classification. Clavien–Dindo classification is widely used throughout surgery for grading adverse events that occur as a result of surgical procedures [17]. Any deviation from the normal postoperative course that did not necessitate pharmacological, surgical, endoscopic, or radiologic intervention was classified as grade 1. Patients of grade 2 were those who require pharmacologic treatment. Patients of grade 3 necessitate surgical, endoscopic, or radiologic intervention. Grade 3a was defined as an intervention performed under regional or local anesthesia, while Grade 3b was defined as an intervention performed under general anesthesia. Life-threatening complications or death received a grade 4 and 5, respectively. Pathological outcomes included tumor, node, metastasis (TNM) stage, histology, number of harvested lymph nodes and positive lymph nodes, tumor size, lymphovascular invasion, and perineural invasion from medical records. Body composition also included phase angle, appendicular skeletal muscle mass (ASM), skeletal muscle index (SMI), and body fluid, intracellular fluid, extracellular fluid, and body fat mass measured using BIA. The eighth edition of the American Joint Committee on Cancer classification system was utilized to identify pathological tumor depth, the number of lymph nodes with metastases, and cancer stage. During each 3-year follow-up examination, a postoperative clinical examination, measurement of serum CEA levels, chest radiography every 3 months, and chest/abdominal CT every 6 months were conducted. After three years, the period between follow-ups was reduced to six months. Recurrence was defined as the radiologically or histologically confirmed existence of tumor. Local recurrence was defined as any tumor recurrence within the surgical field; local recurrence accompanied by synchronous systemic recurrence was considered systemic recurrence. Overall survival (OS) was defined as the interval between the date of surgery and the date of the most recent follow-up visit or the date of death from any cause, whereas disease-free survival (DFS) was defined as the interval between the date of surgery and the date of any recurrence.

### 2.4. Preoperative Evaluation and Surgical Treatment

All patients underwent preoperative evaluations including colonoscopy, computed tomography of the chest and abdomen, and magnetic resonance imaging of the pelvis. Some patients were scanned using positron emission tomography to determine the presence of distant metastases. For CRC, we adhered to the general principles of mesocolic or mesorectal excision and central vessel ligation. The original tumor was removed by performing a precise dissection of the visceral plane from the parietal fascia layer and removing the entire regional mesocolon in one piece.

### 2.5. Bioelectrical Impedance Analysis

A simultaneous multi-frequency impedance measurement equipment with octopolar electrodes, Inbody 770 (Biospace, Seoul, Korea), was used to evaluate the patients’ body composition 1 or 2 weeks before surgery utilizing BIA. The analysis was conducted with individuals in a supine or standing position, wearing light clothing, and with two current and voltage electrodes on each hand and foot (Figure 2). We used 1, 5, 50, 250, 500, and 1000 kHz for the study of intracellular and extracellular water components. The visceral fat area was computed automatically based on trunk impedance, BMI, fat-free mass, fat mass, fat percentage, and muscle mass distribution. We classified the BIA variables as body composition, metabolic index, fat index, muscle index, obesity index, and phase angle. Using Baumgartner’s definition (appendicular/height^2^), the SMI was computed. Sarcopenia was defined as SMI of <7.0 kg/m^2^ in men and <5.7 kg/m^2^ in women using cut-off values in Asian Working Group for Sarcopenia [18].

### 2.6. Assessment of Hematologic Parameters and Inflammation-Based Prognostic Scores

Just prior to surgery, blood samples were drawn from patients as part of preoperative work-up to examine hematologic parameters such as hemoglobin, white blood cell (WBC), hemoglobin, platelet, and albumin. A complete blood cell count was performed on these blood samples to calculate PLR, NLR, PNI, and PIV. The PLR was determined by dividing the absolute number of platelets by the absolute number of lymphocytes. A cut-off value of 150 was utilized to split patients into low and high PLR groups [9]. In addition, Other inflammation-based prognostic scores were also computed (PNI:10× serum albumin concentration (g/dL) + 0.005 × absolute lymphocyte count; NLR: absolute neutrophil count/absolute lymphocyte count). The PIV is a new biomarker that includes neutrophils, lymphocytes, platelets, and monocyte, and preoperative PIV was calculated using the following formula (absolute neutrophil count × platelet count × absolute monocyte count/absolute lymphocyte count) [19].

### 2.7. Statistical Analysis

For continuous outcomes, the findings are provided as means with standard deviation ranges, and for categorical outcomes, as frequencies with percentages. Chi-square and Fisher’s exact tests were used to assess categorical variables. The *t*-test and Mann–Whitney test were used to evaluate continuous variables. A *p*-value of 0.05 or less was regarded as statistically significant. Because of the asymptotic distribution of our data, the optimal cut-off value of visceral fat area (VFA) in our study was estimated using the Contal and O’Quigley method [20]. In survival analysis, the Contal and O’Quigley approach is used to discover cut-off points in continuous variables. The method involves calculating all log-rank statistics and picking the ideal cut point based on the log-rank statistic’s maximum value. This procedure was applied to every conceivable cut-off, and the one with the highest Q statistic was chosen for further examination. Events of the Contal and O’Quigley equations were included in mortality and recurrence. We defined the cut-off values of VFA based on DFS using the Contal and O’Quigley method. VFA ≥ 67.7 cm^2^ was defined as high VFA. Receiver Operating Characteristic (ROC) curves obtained for the visceral fat area measured by bioelectrical impedance analysis (A) and area under the ROC curve (AUC) was 0.538. When we divide two groups using the cut-off values, AUC was 0.606 (B) (Figure 3).

Using the log-rank test for univariate analysis, the Kaplan–Meier method was used to examine the OS and the DFS curve. To determine if adiposity influences DFS, Cox proportional hazards models were utilized. Individual variables’ effects on patient survival were reported as hazard ratios (HRs) with 95% confidence intervals (CIs). The statistical studies were conducted using version 25 of IBM SPSS Statistics (IBM Corp., Armonk, NY, USA) and R Statistical Package (Institute for Statistics and Mathematics, Vienna, Austria, ver. 3.1.2, www.R-project.org, accessed on 23 June 2022).

## 3. Results

### 3.1. Baseline Characteristics of Patients

Based on the cut-off values, 85 (41.7%) patients had low VFA, and 119 (58.3%) patients had high VFA. The patient and tumor characteristics according to low and high adiposity are shown in Table 1. The percentage of men was higher in patients with low VFA than in those with high VFA (77.6% vs. 62.2%; *p* = 0.019). Patients with high VFA showed higher preoperative C-reactive protein (CRP) and BMI than patients with low VFA (0.8 ± 1.7 vs. 0.4 ± 0.7, *p* = 0.047 and 25.0 ± 2.6 vs. 21.3 ± 1.8; *p* < 0.001, respectively). There were no significant differences in age, preoperative CEA level, ASA class, sideness and location of tumor, and distribution of neoadjuvant chemoradiation between the two groups. Immune-inflammatory prognostic markers, including PLR, NLR, PNI, and PIV, showed no significant differences between the two groups.

### 3.2. Perioperative Clinical Outcomes

Table 2 demonstrates that there is no significant difference between the low and high VFA groups in terms of overall perioperative outcomes, including operation time, time to gas out, sips of water, soft food, and hospital stay. There were no statistically significant differences in morbidity 30 days after surgery and the proportion of patients with Clavien-Dindo classification > 3a.

### 3.3. Postoperative Pathologic Outcomes

Table 3 shows postoperative pathologic outcomes. There were no significant differences in tumor and nodal stage, number of retrieved lymph nodes, proportion of patients with more than 12 lymph nodes acquired, number of positive lymph nodes, tumor size, lymphovascular invasion, and perineural invasion between low and high VFA groups. Patients with high VFA showed more moderate and poor differentiation than patients with low VFA (90.6% vs. 83.3% and 6.8% vs. 4.8%; *p* = 0.027).

### 3.4. Body Composition Analysis Using BIA

Table 4 shows the body composition analysis of patients with low and high VFA using BIA. Patients with high VFA had higher weight compared to patients with low VFA (66.2 ± 11.2 vs. 56.4 ± 7.8; *p* = 0.001). Other body components, such as phase angle, appendicular skeletal muscle mass, and skeletal muscle index, did not differ statistically between the two groups. Body fluid, intracellular fluid composition, and extracellular fluid composition did not differ significantly between the two groups; however, the high VFA group had a more body fat mass (20.3 ± 4.8 vs. 11.6 ± 2.6; *p* < 0.001).

### 3.5. Oncologic Outcomes

The median follow-up period was 35.6 months in the low VFA groups and 40.0 months in the high VFA group, without significant differences (Table 5). The high VFA group showed poor prognosis in 5-year OS and DFS, but there were no statistical differences (88.3% vs. 90.3%; *p* = 0.909 and 79.8% vs. 89.3%; *p* = 0.105). There were three cases of recurrence in the low VFA group and fourteen cases of recurrence in the high VFA group. All recurrences were included as systemic recurrence in the low VFA group, but nine cases of systemic recurrence and five cases of local recurrence developed in the high VFA group. In the low VFA group, two patients had liver recurrence and one patient showed peritoneal seeding. Three patients showed liver recurrence, three patients showed lung recurrence, one patient showed bone metastasis and two patients showed peritoneal seeding. Figure 4 shows the relationship between VFA and long-term survival using the Kaplan–Meier curve. OS and DFS were better in patients with low VFA, without statistical differences (OS 90.3% vs. 88.3%; *p* = 0.909, DFS 89.3% vs. 79.8%; *p* = 0.095).

### 3.6. Univariate and Multivariate Survival Analyses of Prognostic Factors

Table 6 showed that univariate analyses revealed that preoperative CRP level, lymph nodal status, perineural invasion, and PIV were significant prognostic factors for OS (Appendix A). Sex, tumor and nodal status, and perineural invasion were identified as significant prognostic factors for DFS (Appendix A). Table 7 showed that multivariate analysis found that preoperative CRP (HR, 3.882; 95% CI, 1.001–15.051; *p* = 0.050) and nodal stage (HR, 7.996; 95% CI, 1.414–45.209; *p* = 0.019) were independent prognostic factors for OS, while sex (HR, 0.110; 95% CI, 0.013–0.905; *p* = 0.040), lymphovascular invasion (HR, 3.560; 95% CI, 1.098–11.544; *p* = 0.034), and VFA (HR, 4.263; 95% CI, 1.280–14.196; *p* = 0.040) were independent prognostic factors for DFS.

## 4. Discussion

This study demonstrated that high visceral fat adiposity preoperatively measured by BIA was associated with higher preoperative CRP levels and poorer histologic differentiation in patients with CRC who underwent curative resection. In the multivariate analysis of oncologic outcomes, visceral fat was an independent prognostic factor for DFS. In contrast, VFA was not significantly linked with short-term clinical and pathological outcomes, immune-inflammatory prognostic indicators, or other body compositions, including skeletal muscle index, body fluid, and phase angle.

Several studies have shown that the operation time is longer, and postoperative complications occur more frequently after surgery in patients with high VFA [10,11]. Visceral obesity was associated with higher surgical difficulty and post-operative morbidity, according to a recent meta-analysis that sought to establish the effect of VFA on laparoscopic CRC surgery [21]. However, another recent study concluded that there was no significant relationship between visceral fat, intraoperative difficulties, and postoperative complications [22]. In this study, there were no significant differences in perioperative short-term outcomes, including total operation time, recovery-related outcomes, or postoperative complications, between patients with low and high VFA. We believe that factors other than visceral obesity had a greater impact on perioperative outcomes in our study. Future research will require further studies, such as multivariate analysis of perioperative outcomes.

CRP is a sensitive indicator of chronic low-grade inflammation, and elevated CRP serum levels have been linked to a variety of diseases including visceral obesity [23,24,25,26,27]. In colorectal cancer, preoperative elevated CRP is a well-known risk factor for recurrence and has poor prognostic value cancers [28]. Previous research has demonstrated that visceral adipocytes contained elevated levels of inflammatory lipid metabolism markers, some of which were associated with CRC tumor stage, and that obesity-induced chronic low-grade inflammation induces oxidative stress factors [11,29]. In the present study, preoperative elevated CRP level was associated with preoperative high VFA that was investigated as an independent poor prognostic factor for DFS. Additionally, preoperative CRP was an independent prognostic factor for OS. Our findings provide clinical evidence for future basic-translational studies on the relationship between visceral obesity, chronic inflammation, and carcinogenesis.

Several studies have produced contradictory findings regarding the clinical significance of visceral fat in relation to oncologic outcomes. Park et al. reported that patients with high VFA showed less lymph node metastasis or lower metastatic lymph node ratio; however, there was no association between VFA and the OS of CRC patients [13]. In contrast, other studies have found a significant association between a high VFA and poor oncologic outcomes [30,31]. In the current study, univariate analysis revealed no statistically significant differences between the high and low VFA groups in terms of oncologic outcomes; however, multivariate analysis revealed the VFA as an independent prognostic factor for DFS. We are cautious in our interpretation of the results, but we interpret them as follows. The percentage of men was higher in patients with low VFA than in those with high VFA (77.6% vs. 62.2%, *p* = 0.019). Meanwhile, men had a statistically significantly lower DFS than women. We think that the prognostic impact of VFA on DFS was offset by female sex being a good prognostic factor in women in the univariate analysis. However, after the gender impact on DFS was regressed through multivariate analysis, VFA was interpreted as being analyzed as an independent prognostic factor for DFS. Based on our results, preoperative VFA could be used as another prognostic factor.

Several investigations have demonstrated the importance of visceral obesity to the development of cancer and the function of omental fat in intraperitoneal carcinogenesis, which was associated with the systemic recurrence of CRC [32,33]. Park et al. showed an association between higher visceral adipocytes and a higher risk of peritoneal seeding in recurrent colorectal cancer [34]. Regarding mechanisms of colorectal cancer development, previous research demonstrated that visceral adipocytes contained elevated levels of inflammatory lipid metabolism markers, some of which were associated with CRC tumor stage, and that obesity-induced chronic low-grade inflammation induces oxidative stress factors [11,29]. 4-Hydroxynonenal (HNE) is the primary result of lipid peroxidation and is responsible for deregulation of several pathways involved in cell proliferation and differentiation, cell survival, apoptosis, and necrosis. The molecular pathways mainly altered by 4-HNE include the mitogen-activated protein kinase, phosphatidylinositol 3-kinase (PI3KCA)/protein kinase B (AKT) signaling pathway, and nuclear factor kappa B (NF-κB). Moreover, accumulation of DNA mutations, such as APC, KRAS, NRAS, BRAF, or PIK3CA, makes obesity a multifactor phenomenon involved in CRC initiation and progression [11].

Numerous studies have investigated the effect of visceral fat composition on the clinical and oncological outcomes of colorectal cancer using dual-energy X-ray absorptiometry or CT [10,11,21]. However, measuring the area of visceral fat using CT or DEXA is a time-consuming task and requires a specific program [35]. In contrast, BIA is a noninvasive, cost-effective, and widely accessible method for nutritional evaluation and anthropometric measurements performed by clinicians and health providers Gupta et al. [36] demonstrated the utility of BIA in the assessment of postoperative malnutrition as a prognostic factor, and Brandstedt et al. [37] demonstrated that males in the highest quartile of body fat percentage had an increased risk for T3/4 tumors and node-positive disease. In addition, several studies have established the validity of evaluating body fat composition with BIA versus CT scans or DEXA, and these studies have demonstrated a high degree of concordance between BIA and CT scan or DEXA. [38,39]. In a prospective cohort research, a high BIA-measured body fat percentage was found to be related with an elevated risk of advanced CRC tumor in men [37]. We expect that research on body fat components and colorectal cancer using BIA will continue.

Nevertheless, our study has some limitations. This study had a retrospective design, which bears the issues of incomplete data and potential selection bias in a single-center study. Although our cut-off values may be adequate for Asian ethnic groups, it may be challenging to apply our findings to other ethnic groups. Additionally, the median follow-up period of patients participating in this study was relatively short (35 months for low VFA and 40 months for high VFA), which limits the analysis of long-term oncological outcomes.

## 5. Conclusions

Preoperative VFA measured by BIA had no significant impact on postoperative clinical outcomes and was an independent prognostic factor for disease-free survival.

## Figures and Tables

**Figure 1 nutrients-14-03971-f001:**
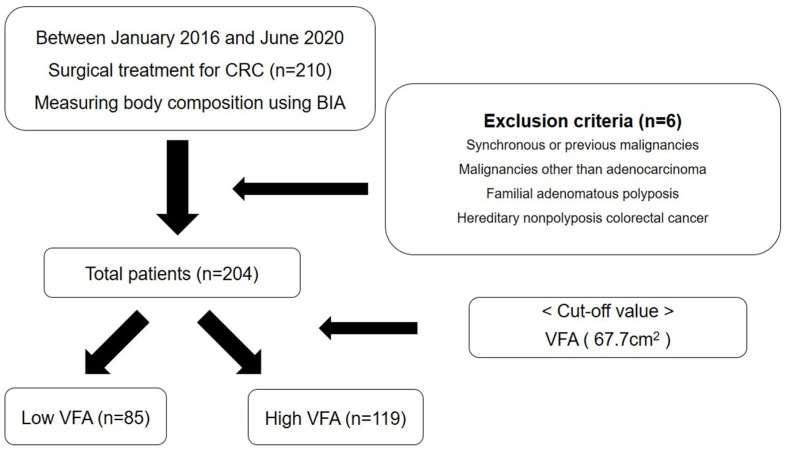
Flow chart of the exclusion criteria.

**Figure 2 nutrients-14-03971-f002:**
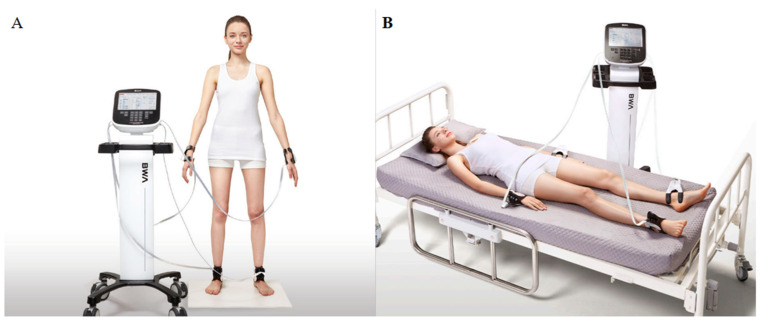
Method of analysis body compositions using Inbody 770. (**A**) Measured by standing position; (**B**) measured by supine position.

**Figure 3 nutrients-14-03971-f003:**
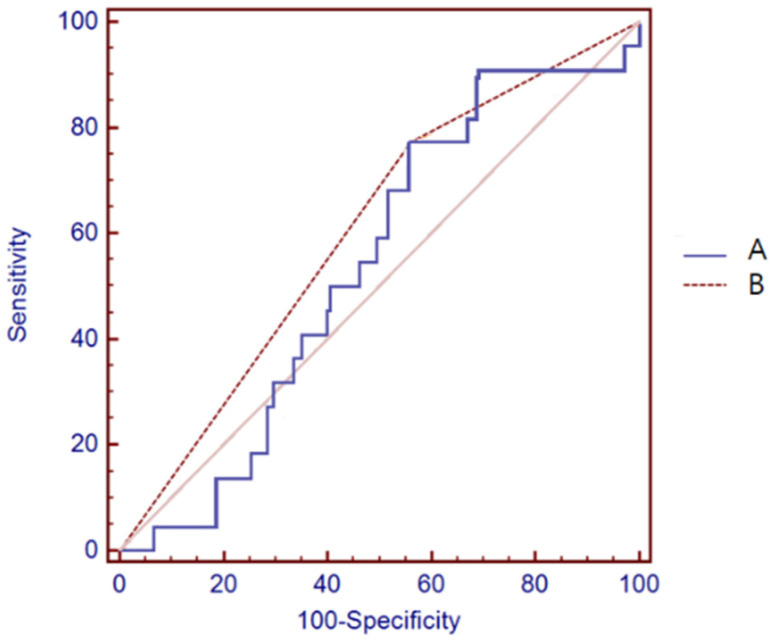
ROC curve for cut-off values of visceral fat area. Calculate all log-rank statistics and optimal cut-off value which is 67.7cm^2^ was selected using largest Q statistic index. Receiver Operating Characteristic (ROC) curves obtained for the visceral fat area measured by bioelectrical impedance analysis (A) and area under the ROC curve (AUC) was 0.538. When we divide two groups using the cut-off values, AUC was 0.606 (B).

**Figure 4 nutrients-14-03971-f004:**
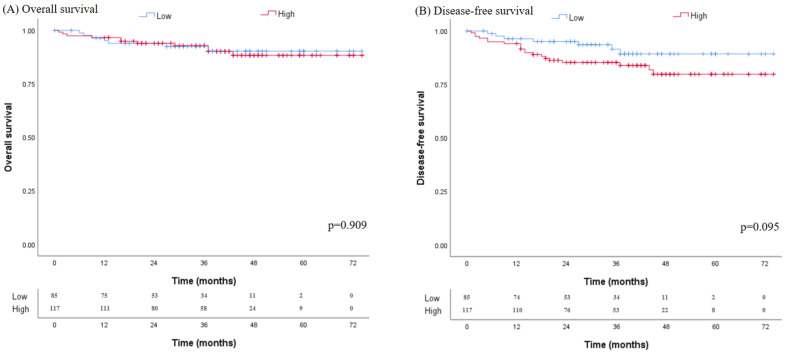
Kaplan–Meier survival curve for the cumulative risk of recurrence. Kaplan–Meier survival curve showed better overall survival (*p* = 0.909) and disease-free survival (*p* = 0.095) without significant difference in patients with low visceral fat area, respectively.

**Table 1 nutrients-14-03971-t001:** Patient and Tumor Characteristics.

	Low VFA (*n* = 85)	High VFA (*n* = 119)	*p* Value
Age (year)	65.9 ± 9.7	66.0 ± 10.2	0.929
Sex			0.019
Male	66 (77.6)	74 (62.2)	
Female	19 (22.4)	45 (37.8)	
Preoperative CEA (ng/mL)	7.0 ± 20.7	5.4 ± 16.0	0.552
Preoperative CRP	0.4 ± 0.7	0.8 ± 1.7	0.047
ASA groups			0.827
I	26 (30.6)	33 (27.7)	
II	49 (57.6)	69 (58.0)	
III	26 (30.6)	33 (27.7)	
BMI (kg/m^2^)	21.3 ± 1.8	25.0 ± 2.6	<0.001
Sideness of tumor			0.599
Right	22 (25.9)	27 (22.7)	
Left	63 (74.1)	92 (77.3)	
Location of tumor			0.740
Colon	43 (50.6)	63 (52.9)	
Rectum	42 (49.4)	56 (47.1)	
Hemoglobin (g/dL)	12.6 ± 2.0	12.4 ± 1.7	0.609
Platelet (×10^3^)	246.2 ± 71.8	241.4 ± 72.3	0.636
WBC (×10^3^)	6.4 ± 2.1	6.0 ± 1.9	0.105
PLR	181.7 ± 114.6	188.2 ± 102.2	0.677
NLR	3.3 ± 3.8	3.1 ± 2.5	0.636
PNI	66.9 ± 27.7	71.2 ± 30.8	0.305
PIV	383.1 ± 710.2	276.9 ± 294.7	0.196
Albumin (g/dL)	4.2 ± 0.5	4.2 ± 0.4	0.603
Neoadjuvant CCRT	17 (20.0)	28 (23.5)	0.549

Categorical variables were analyzed using the chi-square and Fisher’s exact tests. Continuous variables were analyzed using the independent *t*-test and Mann–Whitney U test. Values are presented as mean ± standard deviation or number (%). *p*-value < 0.05 considered as significant. ASA: American society of anesthesiologists; BMI: Body mass index; CEA: Carcinoembryonic antigen; CRP: C-reactive protein; NLR: Neutrophil lymphocyte ratio; PIV: Pan-immune inflammation value; PLR: Platelet-lymphocyte ratio; PNI: Prognostic nutritional index; VFA: Visceral fat area; WBC: White blood cell; CCRT: concurrent chemoradiotherapy.

**Table 2 nutrients-14-03971-t002:** Perioperative Clinical Outcomes.

	Low VFA (*n* = 85)	High VFA (*n* = 119)	*p* Value
Operation time (min)	209.3 ± 112.1	204.0 ± 86.2	0.711
Time to gas out (d)	3.2 ± 2.2	4.0 ± 4.8	0.319
Time to sips of water (d)	4.0 ± 3.1	4.0 ± 4.8	0.983
Time to soft diet (d)	6.3 ± 3.2	6.6 ± 5.1	0.603
Time to hospital stay (d)	10.4 ± 6.4	10.2 ± 6.2	0.773
Morbidity within 30 days after surgery	28 (32.9)	40 (33.6)	0.920
Clavien–Dindo classifications > 3a	17 (20.0)	25 (21.0)	0.861

Categorical variables were analyzed using the chi-square and Fisher’s exact tests. Continuous variables were analyzed using the independent *t*-test and Mann–Whitney U test. Values are presented as mean ± standard deviation or number (%). *p*-value < 0.05 considered as significant. d: day; min: minute; VFA: visceral fat area.

**Table 3 nutrients-14-03971-t003:** Postoperative Pathologic Outcomes.

	Low VFA (*n* = 85)	High VFA (*n* = 119)	*p* Value
Tumor stage			0.114
T1	16 (18.8)	33 (24.0)	
T2	16 (18.8)	42 (20.6)	
T3	43 (50.6)	99 (48.5)	
T4	10 (11.8)	14 (6.9)	
Nodal stage			0.945
N0	55 (64.7)	79 (66.4)	
N1	21 (24.7)	27 (22.7)	
N2	9 (10.6)	13 (10.9)	
Histology			0.027
Well differentiated	10 (11.9)	3 (2.6)	
Moderately differentiated	70 (83.3)	106 (90.6)	
Poorly differentiated	4 (4.8)	8 (6.8)	
Retrieved LNs	19.5 ± 9.4	18.1 ± 9.2	0.310
LN > 12	77 (90.6)	99 (83.2)	0.130
Positive LNs	1.0 ± 2.0	0.9 ± 2.1	0.807
Tumor size (cm)	3.9 ± 2.1	3.5 ± 2.1	0.211
Lymphovascular invasion	27 (31.8)	27 (23.5)	0.192
Perineural invasion	16 (19.3)	25 (22.5)	0.584

Categorical variables were analyzed using the chi-square and Fisher’s exact tests. Continuous variables were analyzed using the independent *t*-test and Mann–Whitney U test. Values are presented as mean ± standard deviation or number (%). *p*-value < 0.05 considered as significant. LN: Lymph node; VFA: Visceral fat area.

**Table 4 nutrients-14-03971-t004:** Inbody 770 Body Composition Analysis of Patients.

	Low VFA (*n* = 85)	High VFA (*n* = 119)	*p* Value
Height (cm)	162.3 ± 8.6	162.4 ± 9.5	0.980
Weight (kg)	56.4 ± 7.8	66.2 ± 11.2	<0.001
Phase angle (′)	5.1 ± 0.6	5.0 ± 0.7	0.629
ASM (kg)	7.0 ± 1.1	7.1 ± 1.1	0.650
SMI (kg/m^2^)	2.7 ± 0.5	2.7 ± 0.4	0.749
Body fluid	33.1 ± 5.3	33.9 ± 6.7	0.347
ICF (%)	20.3 ± 3.4	20.8 ± 4.2	0.362
ECF (%)	12.8 ± 2.0	13.1 ± 2.6	0.328
BFM (kg)	11.6 ± 2.6	20.3 ± 4.8	<0.001

Values are presented as mean ± standard deviation or number (%). *p*-value < 0.05 considered as significant. ASM: Appendicular skeletal muscle mass; BFM = Body fat mass; ECF: Extracellular fluid; ICF: Intracellular fluid; SMI: Skeletal muscle index; VFA: Visceral fat area.

**Table 5 nutrients-14-03971-t005:** Oncologic Outcomes.

	Low VFA (*n* = 85)	High VFA (*n* = 119)	*p* Value
Median follow-up (months)	35.6 ± 16.2	40.0 ± 18.0	0.073
5 yr OS (%)	90.3	88.3	0.909
5 yr DFS (%)	89.3	79.8	0.105
Recurrence	3	14	
Recurrence pattern			0.070
Systemic recurrence	3	9	
Local recurrence	0	5	

Categorical variables were analyzed using the chi-square and Fisher’s exact tests. Continuous variables were analyzed using the independent *t*-test and Mann–Whitney U test. Values are presented as mean ± standard deviation or number (%). *p*-value < 0.05 considered as significant. DFS: disease free survival; OS: Overall survival; VFA: Visceral fat area.

**Table 6 nutrients-14-03971-t006:** Prognostic Factors of 5-year Survival by Univariate Analysis.

Prognostic Factor	N	OS (5 Years, %)	Log Rank *p*-Value	DFS (5 Years, %)	Log Rank *p*-Value
Visceral fat area			0.909		0.105
Low	85	90.3		89.3	
High	119	88.3		79.8	
Age			0.689		0.917
≤65	89	90.2		84.7	
>65	115	87.8		82.1	
Sex			0.060		0.016
Male	140	85.5		79.5	
Female	64	96.9		92.0	
BMI			0.332		0.327
High (>25)	52	92.8		90.2	
Low (<25)	152	87.5		80.8	
ASA score			0.253		0.571
1	59	94.9		81.9	
2 and 3	145	86.6		84.0	
Sideness			0.431		0.687
Right sided	49	84.2		79.4	
Left sided	155	90.6		84.7	
Pre-op CEA (ng/mL)			0.164		0.072
<5	162	90.6		85.0	
≥5	42	82.2		76.7	
Pre-op CRP (mg/L)			0.043		0.623
<0.3	99	90.0		86.6	
≥0.3	55	80.3		83.7	
Tumor stage			0.119		0.037
T1 and T2	91	92.8		92.0	
T3 and T4	113	85.6		76.0	
Nodal stage			<0.001		0.001
Nodal negative	133	94.5		90.4	
Nodal positive	71	79.0		69.5	
Differentiation			0.822		0.488
Well	15	92.9		92.9	
Moderate and poor	188	89.1		83.0	
Lymphovascular invasion			0.085		0.089
No	146	90.8		84.8	
Yes	54	83.3		78.3	
Perineural invasion			0.030		0.004
No	153	92.1		85.5	
Yes	41	80.6		72.6	
LN harvest			0.314		0.363
≥12	176	88.3		82.2	
<12	28	92.3		92.9	
PIV			0.010		0.298
Low	145	94.1		86.1	
High	59	77.3		77.1	
Phase angle			0.215		0.944
Low	117	92.1		85.3	
High	87	84.3		82.4	
Sarcopenia			0.311		0.313
No	143	90.3		85.0	
Yes	61	85.6		79.4	

Kaplan–Meier method with the log-rank test is used for univariate analysis. *p*-value < 0.05 considered as significant. ASA: American society of anesthesiologists; BMI: Body mass index; CEA: Carcinoembryonic antigen; CRP: C-reactive protein; LN: Lymph node; PIV: Pan-immune inflammation value.

**Table 7 nutrients-14-03971-t007:** Prognostic Factors of Overall Survival and Disease-Free Survival in Multivariate Analysis.

Variables	Reference Category	Overall Survival		Disease-Free Survival	
		HR (95% CI)	*p*-Value	HR (95% CI)	*p*-Value
VFA					
High	Low	1.67 (0.50–5.56)	0.401	4.26 (1.28–14.20)	0.018
Sex					
Female	Male	0.59 (0.12–2.87)	0.509	0.11 (0.01–0.91)	0.040
Sarcopenia					
Yes	No	1.57 (0.49–5.08)	0.451	2.31 (0.79–6.77)	0.126
Pre-OP CEA					
≥5	<5	0.96 (0.29–3.16)	0.942	0.92 (0.28–3.04)	0.890
CRP					
≥0.3	<0.3	3.88 (1.00–15.05)	0.050	1.38 (0.44–4.35)	0.585
PIV					
High	Low	1.17 (0.316–4.356)	0.811	0.62 (0.19–2.03)	0.426
Tumor stage					
T3, T4	T1, T2	0.91 (0.14–6.08)	0.926	1.11 (0.27–4.63)	0.889
Nodal stage					
N1, N2	N0	8.00 (1.41–45.21)	0.019	1.28 (0.37–4.45)	0.702
Lymphovascular invasion					
Yes	No	3.06 (0.88–10.63)	0.078	3.56 (1.10–11.54)	0.034
Perineural invasion					
Yes	No	1.10 (0.31–3.95)	0.880	2.46 (0.73–8.25)	0.144

Cox proportional hazard models were used for statistical analysis. The effects of individual variables on patient survival were expressed as hazard ratios (HRs) with 95% confidence intervals (CIs). *p*-value < 0.05 considered as significant. CEA: Carcinoembryonic antigen; CRP: C-reactive protein; PIV: Pan-immune inflammation value; VFA: Visceral fat area.

## Data Availability

The data presented in this study are available on request from the corresponding author. The data are not publicly available due to the patient’s privacy including this study.

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
