# Peer review of "Impact of Preoperative Visceral Fat Area Measured by Bioelectrical Impedance Analysis on Clinical and Oncologic Outcomes of Colorectal Cancer"

_nutrients, 2022, doi:10.3390/nu14193971_

Round 1

Reviewer 1 Report

The authors conducted a retrospective analysis regarding to the prognostic roles of visceral fat area on survival of colorectal cancer patients. This is an interesting study and proves outcomes participation for physicians who treated patients with colorectal cancer. However, there are some issues warranted further explanations.

1. The authors used bioelectrical impedance analysis to estimate the visceral fat area. Given that readers might not be familiar with bioelectrical impedance analysis, the authors should provide brief introduction the mechanism of bioelectrical impedance analysis. Furthermore, was bioelectrical impedance analysis a reliable tool to measure visceral fat area? The authors should provide the evidences regarding the validation of bioelectrical impedance analysis with visceral fat area.

2. The cut-off value of visceral fat area was determined by ROC curve. Please provided the figure of ROC curve and c-index. 

3. The authors stated in “introduction” that our study aimed to compare the impact of visceral fat measured using BIA on clinical, pathologic, and oncologic outcomes in patients who underwent surgical treatment for CRC. However, the authors compared all inbody 770 body composition analysis of patients. Would you please describe the all purposes of your study? I wonder that why you want to present all body composition analysis data from BIA.                                                                                                                                                                                                                                                                                                                                                                                                                                                                  

4. Table 1 should provide the clinical stage or pathologic stage of patients, as well as the distribution of neoadjuvant chemoradiotherapy and adjuvant chemotherapy. The primary tumor location should further classify into colon and rectum because the treatment and prognosis were totally different between colon cancer and rectal cancer.

5. Table 6 and 7 used sarcopenia as a parameter. However, the definition of sarcopenia was not present in the context. Please add the definition of sarcopenia in “methods and material”

6. According to the results, the high VFA was not associated with perioperative outcomes, pathologic outcomes, disease free survival and overall survival. Thus, I think that there were no clinical implications regarding visceral fat area for colorectal cancer patients treated with radical surgery. The authors should discuss the clinical roles of visceral fat area in colorectal cancer patients, as well as clinical utility of this parameter for outcomes prediction. Moreover, according to table 5 and table 6, the 5-year disease free survival was not significant between high VFA and low VFA. However, high VFA was significantly associated with disease free survival in multivariate analysis in Table 7. It is not reasonable. Please check your data carefully.

7. Figures are all missing. Please correct your manuscript. 

Author Response

Thank you for your kindly review for our study.

We take your comments and revise the manuscript. But there is no method to upload the figure before the replying to comments. So We include the figures in manuscript. Please see the attachment.

  1. The authors used bioelectrical impedance analysis to estimate the visceral fat area. Given that readers might not be familiar with bioelectrical impedance analysis, the authors should provide brief introduction the mechanism of bioelectrical impedance analysis. Furthermore, was bioelectrical impedance analysis a reliable tool to measure visceral fat area? The authors should provide the evidences regarding the validation of bioelectrical impedance analysis with visceral fat area.

Response: We agree with your comment that readers might not be familiar with bioelectrical impedance analysis. As such, we have added a brief introduction to the mechanism of bioelectrical impedance analysis in the bioelectrical impedance analysis section as well as in Figure 2.

  “BIA was performed using a simultaneous multi-frequency impedance measurement device with octapolar electrodes, Inbody 770 (Biospace, Republic of Korea), to estimate the patients’ body composition 1 week or 2 weeks prior to surgery. The analysis was performed with patients wearing light clothing and in a supine or standing position with two current and voltage electrodes on each hand and foot (Figure 2). The frequencies used for the analysis of intracellular and extracellular water compositions were 1, 5, 50, 250, 500, and 1000 kHz. The visceral fat area was automatically calculated from the trunk impedance, BMI, fat-free mass, fat mass, fat percentage, and distribution of muscle mass. Among the various BIA parameters, we categorized the variables as either body composition, metabolic index, fat index, muscle index, obesity index, and phase angle. The SMI was calculated using Baumgartner’s definition (appendicular/height2). Sarcopenia was defined as SMI <7.0 kg/m2 in men and <5.7 kg/m2 in women using cut-off values defined by the Asian Working Group for Sarcopenia [1].”

  Regarding bioelectrical impedance analysis as a reliable tool to measure visceral fat area, we have provided evidence regarding the validation of bioelectrical impedance analysis with visceral fat area in the Discussion section.

  “Gupta et al.[2] demonstrated the utility of BIA in the assessment of postoperative malnutrition as a prognostic factor, while Brandstedt et al.[3] demonstrated that males in the highest quartile of body fat percentage had an increased risk of T3/4 tumors and node-positive disease. In addition, several studies have previously established the validity of evaluating body fat composition with BIA versus CT scans or DEXA, and have demonstrated a high degree of concordance between BIA and CT scan or DEXA[4,5].”

  1. The cut-off value of visceral fat area was determined by ROC curve. Please provided the figure of ROC curve and c-index. 

Response: We thank the reviewer for their comments. We consulted a statistician to determine the optimal cutoff value of visceral fat area in this study, who recommended applying the Contal and O'Quigley method over the ROC curve; indeed, because of the asymptotic distribution of our data, the optimal cutoff value of visceral fat area (VFA) in our study was estimated using the Contal and O’Quigley method. (https://doi.org/10.1016/S0167-9473(98)00096-6)

  Furthermore, the Contal and O'Quigley method was used to identify cut-off points for continuous variables in the survival analysis. We calculated all log-rank statistics and selected the optimal cutoff point based on the maximization of the log-rank statistic, as mentioned in the article.

  We have added Figure 3 and the following sentence to explain this: “Receiver operating characteristic (ROC) curves obtained for visceral fat area measured by bioelectrical impedance analysis (A) and area under the ROC curve (AUC) were 0.538. When we divided the two groups using the defined cutoff values, the AUC was 0.606 (B) (Figure 3).”

  1. The authors stated in “introduction” that our study aimed to compare the impact of visceral fat measured using BIA on clinical, pathologic, and oncologic outcomes in patients who underwent surgical treatment for CRC. However, the authors compared all inbody 770 body composition analysis of patients. Would you please describe the all purposes of your study? I wonder that why you want to present all body composition analysis data from BIA.                          

Response: We thank the reviewer for their comments. Our study aimed to investigate the impact of preoperative visceral fat area measured using bioelectrical impedance analysis on the clinical and oncologic outcomes of colorectal cancer. To determine whether this factor affects postoperative clinical and oncologic outcomes, we calculated the cut-off VFA at the point that had the most significant effect on oncologic outcome, and demographic characteristics of the two groups, body composition, postoperative clinical outcomes, and oncologic outcomes were analyzed. We have modified the sentence describing the purpose of this study based on the reviewer’s comments.

  ‘The purpose of our study was to investigate the impact of preoperative visceral fat area measured using bioelectrical impedance analysis on the oncologic outcomes of colorectal cancer.

  Regarding body composition analysis data from BIA, many studies have previously investigated whether a single body composition affects oncological outcomes. However, body components that may be correlated can influence oncological outcomes. For example, obese patients generally have less sarcopenia, which is sometimes referred to as the "obesity paradox". Therefore, we investigated the differences in other body compositions based on visceral obesity and conducted a multivariate analysis using multiple variables, including data on other body compositions that can have clinical or oncological effects.

  1. Table 1 should provide the clinical stage or pathologic stage of patients, as well as the distribution of neoadjuvant chemoradiotherapy and adjuvant chemotherapy. The primary tumor location should further classify into colon and rectum because the treatment and prognosis were totally different between colon cancer and rectal cancer.

Response: Thank you for your suggestion. In accordance with your suggestion, the pathologic stage of patients is described in table 3, and the distribution of neoadjuvant chemoradiation and the classification of the primary tumor site as colon or rectum have been added to Table 1.  

  1. Table 6 and 7 used sarcopenia as a parameter. However, the definition of sarcopenia was not present in the context. Please add the definition of sarcopenia in “methods and material”

Response: We agree with your comment. We have added a definition for sarcopenia to the Materials and Methods section.

  “Sarcopenia was defined as an SMI of <7.0 kg/m2 in men and <5.7 kg/m2 in women using the cut-off values defined by the Asian Working Group for Sarcopenia.”

  1. According to the results, the high VFA was not associated with perioperative outcomes, pathologic outcomes, disease free survival and overall survival. Thus, I think that there were no clinical implications regarding visceral fat area for colorectal cancer patients treated with radical surgery. The authors should discuss the clinical roles of visceral fat area in colorectal cancer patients, as well as clinical utility of this parameter for outcomes prediction. Moreover, according to table 5 and table 6, the 5-year disease free survival was not significant between high VFA and low VFA. However, high VFA was significantly associated with disease free survival in multivariate analysis in Table 7. It is not reasonable. Please check your data carefully.

Response: We agree with your comments on the need to discuss the clinical role of visceral fat area in colorectal cancer patients, as well as the clinical utility of this parameter for outcome prediction. In our study, we determined that preoperative VFA measured by BIA was an independent prognostic factor for disease-free survival, although it had no effect on postoperative clinical outcomes. Regarding our survival data, although this was a retrospective study, it was conducted using a prospectively collected database that was systematically managed. We carefully analyzed the data again and found no errors. The corresponding text in the Discussion section has been revised in accordance with the reviewer’s suggestions.

We have cautiously interpreted our results as follows: the percentage of men was higher in patients with a low VFA than in those with a high VFA (77.6% vs. 62.2%; p = 0.019). Meanwhile, men had a significantly lower DFS than women. We believe that the prognostic impact of VFA on DFS was offset in women in the univariate analysis as female sex is a good prognostic factor. However, after the sex impact on DFS was regressed through multivariate analysis, VFA was interpreted as being analyzed as an independent prognostic factor for DFS. Based on our results, preoperative VFA can be used as a prognostic factor.

  1. Figures are all missing. Please correct your manuscript.

Response: We sincerely apologize for this oversight. We have checked the figures and have uploaded the manuscript again

< References >

  1. Chen, L.K.; Woo, J.; Assantachai, P.; Auyeung, T.W.; Chou, M.Y.; Iijima, K.; Jang, H.C.; Kang, L.; Kim, M.; Kim, S.; et al. Asian Working Group for Sarcopenia: 2019 Consensus Update on Sarcopenia Diagnosis and Treatment. J Am Med Dir Assoc 2020, 21, 300-307 e302, doi:10.1016/j.jamda.2019.12.012.
  2. Gupta, D.; Lis, C.G.; Dahlk, S.L.; King, J.; Vashi, P.G.; Grutsch, J.F.; Lammersfeld, C.A. The relationship between bioelectrical impedance phase angle and subjective global assessment in advanced colorectal cancer. Nutr J 2008, 7, 19, doi:10.1186/1475-2891-7-19.
  3. Brandstedt, J.; Wangefjord, S.; Nodin, B.; Gaber, A.; Manjer, J.; Jirstrom, K. Gender, anthropometric factors and risk of colorectal cancer with particular reference to tumour location and TNM stage: a cohort study. Biol Sex Differ 2012, 3, 23, doi:10.1186/2042-6410-3-23.
  4. Kim, M.; Shinkai, S.; Murayama, H.; Mori, S. Comparison of segmental multifrequency bioelectrical impedance analysis with dual-energy X-ray absorptiometry for the assessment of body composition in a community-dwelling older population. Geriatr Gerontol Int 2015, 15, 1013-1022, doi:10.1111/ggi.12384.
  5. Lee, D.H.; Park, K.S.; Ahn, S.; Ku, E.J.; Jung, K.Y.; Kim, Y.J.; Kim, K.M.; Moon, J.H.; Choi, S.H.; Park, K.S.; et al. Comparison of Abdominal Visceral Adipose Tissue Area Measured by Computed Tomography with That Estimated by Bioelectrical Impedance Analysis Method in Korean Subjects. Nutrients 2015, 7, 10513-10524, doi:10.3390/nu7125548.

Reviewer 2 Report

Dear colleagues, I have reviewed the presented work, Prognostic impact of visceral fat area measured by bioelectrical impedance analysis on oncologic outcomes of colorectal cancer. Which aims to compare the impact of visceral fat measured with BIA on clinical, pathological and oncological outcomes in patients who underwent surgical treatment for CRC.

Of which I have the following observations:

1. Given the impact of BIA analysis on the submitted work, this reviewer believes that the method should be described in more detail (lines 99-104). Such as the parameters used to calculate each of the aforementioned indices, the data that is entered into the Biospace that is not determined, the conditions in which the patients were measured, reference values used, etc.

2. Indicate in the methodology how the Visceral Fat Area was calculated.

3. Authors are requested to follow the guidelines suggested for the presentation of the manuscript, where it is clearly indicated that the figures and tables must be inserted in the text, after mentioning them.

4. In the methodology, it is recommended to indicate the reference parameters used in the results, for example, what do the ASA scores 1, 2 and 3 mean clinically?

5. It is recommended that the results tables include the p value and the statistical analysis used to facilitate understanding of the study.

6. Line 216 indicates that CRP has a high prognostic value for CRC and attributes it directly to chronic inflammation due to CRC (line 220). However, CRP is also increased in obesity (https://doi.org/10.1111/obr.12003) How do you explain this?

7. The conclusion is not clear and does not align with the title of the work

8. The title of the work does not seem appropriate to this reviewer, according to the findings of the work presented.

Author Response

Thank you for your kindly review for our study.

We take your comments and revise the manuscript. But there is no method to upload the figure before the replying to comments. So we include the figures in manuscript. Please see the attachment.

  1. Given the impact of BIA analysis on the submitted work, this reviewer believes that the method should be described in more detail (lines 99-104). Such as the parameters used to calculate each of the aforementioned indices, the data that is entered into the Biospace that is not determined, the conditions in which the patients were measured, reference values used, etc.

Response: Thank you for your suggestion. We have added a brief introduction to the mechanism of bioelectrical impedance analysis and the method to calculate each of the mentioned indices and conditions in which the patients were measured in the “Bioelectrical impedance analysis section” and Figure 2 based on the reviewer’s comments. Regarding the cut-off values of body composition parameters, including skeletal muscle index, body fat, body fluid, and phase angle, there are no optimal reference values for each index except the skeletal muscle index according to a review of the literature.

  “BIA was performed using a simultaneous multi-frequency impedance measurement device with octapolar electrodes, the Inbody 770 (Biospace, Republic of Korea), to estimate the patients’ body composition 1 or 2 weeks before the surgery. The analysis was performed with patients wearing light clothing in a supine or standing position with two current and voltage electrodes on each hand and foot (Figure 2). The frequencies used for the analysis of intracellular and extracellular water compositions were 1, 5, 50, 250, 500, and 1000 kHz. The visceral fat area was automatically calculated from the trunk impedance, BMI, fat-free mass, fat mass, fat percentage, and distribution of muscle mass. Among the various BIA parameters, we categorized the variables as body composition, metabolic index, fat index, muscle index, obesity index, and phase angle. The SMI was calculated using Baumgartner’s definition (appendicular/height2). Sarcopenia was defined as an SMI <7.0 kg/m2 in men and <5.7 kg/m2 in women using cut-off values from the Asian Working Group for Sarcopenia[1].”

  1. Indicate in the methodology how the Visceral Fat Area was calculated.

Response: We agree with this comment. We have added a description of the methodology to calculate the visceral fat area in the following sentences.

  “Visceral fat area was automatically calculated from trunk impedance, BMI, fat-free mass, fat mass, fat percentage, and the distribution of muscle mass.”

  1. Authors are requested to follow the guidelines suggested for the presentation of the manuscript, where it is clearly indicated that the figures and tables must be inserted in the text, after mentioning them.

Response: Thank you for your comment. We have rechecked the figure and tables and marked those that have been inserted in the text.

  1. In the methodology, it is recommended to indicate the reference parameters used in the results, for example, what do the ASA scores 1, 2 and 3 mean clinically?

Response: Thank you for your comment. We have not included a description of the clinical significance of reference parameters used in the results, including ASA score and Clavien-Dindo classification, to the Methodology section of this text.

  The ASA score is used to assess and communicate a patient’s pre-anesthesia medical comorbidities. ASA I describes healthy, normal patients. As the ASA score increases, the severity of comorbidity increases, and ASA III is used to describe patients with severe systemic disease [2].”

The Clavien-Dindo classification is widely used to grade adverse events that occur as a result of numerous surgical procedures [3]. Any deviation from the normal postoperative course that did not necessitate pharmacological, surgical, endoscopic, or radiological intervention was classified as grade 1. Grade 2 patients required pharmacological treatment. Grade 3 patients require surgical, endoscopic, or radiological interventions. Grade 3a was defined as an intervention performed under regional or local anesthesia, while 3b was defined as an intervention performed under general anesthesia. Life-threatening complications and death received grades of 4 and 5, respectively.”

  1. It is recommended that the results tables include the p value and the statistical analysis used to facilitate understanding of the study.

Response: Thank you for your comment. We have now added the p-value and the statistical analysis used in each table.

  1. Line 216 indicates that CRP has a high prognostic value for CRC and attributes it directly to chronic inflammation due to CRC (line 220). However, CRP is also increased in obesity (.org/10.1111/obr.12003) How do you explain this?

Response: We agree with your comment that CRP level is also increased in obesity. Previous studies have reported that CRP is a sensitive indicator of chronic low-grade inflammation, and that elevated CRP serum levels have been linked to various diseases, including visceral obesity. Moreover, we suggest clinical evidence for future basic-translational studies on the relationship between visceral obesity, chronic inflammation, and carcinogenesis. The modified content is described in the discussion section.

  “CRP is a sensitive indicator of chronic low-grade inflammation, and elevated CRP serum levels have been linked to a variety of diseases, including visceral obesity[4-8]. In colorectal cancer, an elevated preoperative CRP level is a well-known risk factor for recurrence, with poor prognostic value [9]. Previous studies have demonstrated that visceral adipocytes contain elevated levels of inflammatory lipid metabolism markers, some of which are associated with the CRC tumor stage, and that obesity-induced chronic low-grade inflammation induces oxidative stress factors[10,11]. In the present study, preoperative elevated CRP levels were associated with preoperative high VFA, which was identified as an independent poor prognostic factor for DFS. Additionally, preoperative CRP level was an independent prognostic factor for OS. Our findings provide clinical evidence for future basic-translational studies on the relationship between visceral obesity, chronic inflammation, and carcinogenesis.”

  1. The conclusion is not clear and does not align with the title of the work

Response: We agree with your comments and have revised the conclusion to align with the title of the work, as follows:

  “Preoperative VFA measured using BIA had no significant impact on postoperative clinical outcomes, and was an independent prognostic factor for disease-free survival.”

  1. The title of the work does not seem appropriate to this reviewer, according to the findings of the work presented.

Response: We agree with your comment that the title of the work does not seem appropriate according to the findings of the work presented. To determine whether preoperative visceral fat area measured by bioelectrical impedance analysis affects postoperative clinical and oncologic outcomes, we identified the cut-off VFA with the most significant effect on oncologic outcome, and demographic characteristics of the two groups, body composition, postoperative clinical outcomes, and oncologic outcomes were analyzed. The title has been revised in accordance with the reviewer's recommendations.

  “The Impact of preoperative visceral fat area measured by bioelectrical impedance analysis on clinical and oncologic outcomes of colorectal cancer”

< References >

  1. Chen, L.K.; Woo, J.; Assantachai, P.; Auyeung, T.W.; Chou, M.Y.; Iijima, K.; Jang, H.C.; Kang, L.; Kim, M.; Kim, S.; et al. Asian Working Group for Sarcopenia: 2019 Consensus Update on Sarcopenia Diagnosis and Treatment. J Am Med Dir Assoc 2020, 21, 300-307 e302, doi:10.1016/j.jamda.2019.12.012.
  2. Hurwitz, E.E.; Simon, M.; Vinta, S.R.; Zehm, C.F.; Shabot, S.M.; Minhajuddin, A.; Abouleish, A.E. Adding Examples to the ASA-Physical Status Classification Improves Correct Assignment to Patients. Anesthesiology 2017, 126, 614-622, doi:10.1097/ALN.0000000000001541.
  3. Dindo, D.; Demartines, N.; Clavien, P.A. Classification of surgical complications: a new proposal with evaluation in a cohort of 6336 patients and results of a survey. Ann Surg 2004, 240, 205-213, doi:10.1097/01.sla.0000133083.54934.ae.
  4. Saijo, Y.; Kiyota, N.; Kawasaki, Y.; Miyazaki, Y.; Kashimura, J.; Fukuda, M.; Kishi, R. Relationship between C-reactive protein and visceral adipose tissue in healthy Japanese subjects. Diabetes Obes Metab 2004, 6, 249-258, doi:10.1111/j.1462-8902.2003.0342.x.
  5. Tsuriya, D.; Morita, H.; Morioka, T.; Takahashi, N.; Ito, T.; Oki, Y.; Nakamura, H. Significant correlation between visceral adiposity and high-sensitivity C-reactive protein (hs-CRP) in Japanese subjects. Intern Med 2011, 50, 2767-2773, doi:10.2169/internalmedicine.50.5908.
  6. Danesh, J.; Wheeler, J.G.; Hirschfield, G.M.; Eda, S.; Eiriksdottir, G.; Rumley, A.; Lowe, G.D.; Pepys, M.B.; Gudnason, V. C-reactive protein and other circulating markers of inflammation in the prediction of coronary heart disease. The New England journal of medicine 2004, 350, 1387-1397, doi:10.1056/NEJMoa032804.
  7. Dehghan, A.; Kardys, I.; de Maat, M.P.; Uitterlinden, A.G.; Sijbrands, E.J.; Bootsma, A.H.; Stijnen, T.; Hofman, A.; Schram, M.T.; Witteman, J.C. Genetic variation, C-reactive protein levels, and incidence of diabetes. Diabetes 2007, 56, 872-878, doi:10.2337/db06-0922.
  8. Santa-Paavola, R.; Lehtinen-Jacks, S.; Jaaskelainen, T.; Mannisto, S.; Lundqvist, A. The association of high-sensitivity C-reactive protein with future weight gain in adults. International journal of obesity 2022, 46, 1234-1240, doi:10.1038/s41366-022-01101-7.
  9. Liao, C.K.; Yu, Y.L.; Lin, Y.C.; Hsu, Y.J.; Chern, Y.J.; Chiang, J.M.; You, J.F. Prognostic value of the C-reactive protein to albumin ratio in colorectal cancer: an updated systematic review and meta-analysis. World J Surg Oncol 2021, 19, 139, doi:10.1186/s12957-021-02253-y.
  10. Liesenfeld, D.B.; Grapov, D.; Fahrmann, J.F.; Salou, M.; Scherer, D.; Toth, R.; Habermann, N.; Bohm, J.; Schrotz-King, P.; Gigic, B.; et al. Metabolomics and transcriptomics identify pathway differences between visceral and subcutaneous adipose tissue in colorectal cancer patients: the ColoCare study. Am J Clin Nutr 2015, 102, 433-443, doi:10.3945/ajcn.114.103804.
  11. Martinez-Useros, J.; Garcia-Foncillas, J. Obesity and colorectal cancer: molecular features of adipose tissue. J Transl Med 2016, 14, 21, doi:10.1186/s12967-016-0772-5.

Reviewer 3 Report

NIL

Author Response

Thank you for your kindly review for our study.

We take your comments and revise the manuscript. But there is no method to upload the figure before the replying to comments. So we include the figures in manuscript. Please see the attachment.

  1. Usage of and twice in line-11 of page 1 as in “ …relationship between visceral fat area (VFA), clinical, and oncologic outcomes in CRC” rather than “… relationship between visceral fat area (VFA) and clinical, and oncologic outcomes in CRC.”

Response: Thank you for your suggestion. We agree that the meaning of the sentence was unclear, and have therefore modified it accordingly.

  “This study aimed to investigate the impact of preoperative visceral fat area measured by bioelectrical impedance analysis on the clinical and oncologic outcomes of colorectal cancer.”

  1. “…Controversies exists…” Rather than “…Controversies exist…” in line – 37 of page – 1.

Response: We have revised the manuscript according to the reviewer's comment.

  “Controversies exists regarding the correlation between visceral obesity and colon cancer outcomes.”

3.In the sentence, “Previous studies have shown the relationships between body composition, including sarcopenia, using skeletal muscle index, visceral fat, phase angle, and clinical and oncologic outcomes of CRC” of lines 48-50 of Page – 2 where, use of the word Sarcopenia needs to be justified since it is a condition and it may not be included. 

Response: We agree with your comments that the use of the word sarcopenia needs to be justified because it is a condition and have revised the manuscript accordingly.

 “Previous studies have shown the relationships between body composition, including skeletal muscle index, visceral fat, phase angle, and clinical and oncologic outcomes of CRC.”

Round 2

Reviewer 1 Report

The author had revised their manuscript according to our suggestion. However, the figures are still missing and the order of figures are wrong. Please correct it.

Reviewer 2 Report

This reviewer considers that the modifications made to the work adequately reflect the results and conclusions of the same; however, authors are asked to review the manuscript submission format again; since in the template it indicates that the figures and tables should be placed just after where they were mentioned and in the revised document these elements are found at the end of it, which does not facilitate the analysis.